# An Evaluation of Paddy Rice as an Alternative Energy Source in Protein-Restricted Diets for Growing, Early-Finishing, and Late-Finishing Pigs

**DOI:** 10.3390/ani14030391

**Published:** 2024-01-25

**Authors:** Zijuan Wu, Wenli Li, Huixia Wang, Yali Li

**Affiliations:** Hunan Provincial Key Laboratory of Animal Intestinal Function and Regulation, Hunan Normal University, Changsha 410081, China; wuzijuan2000@163.com (Z.W.); m18273887463@163.com (W.L.); 13873499432@163.com (H.W.)

**Keywords:** alternatives, paddy rice, growth performance, growing–finishing pigs

## Abstract

**Simple Summary:**

There is a need to explore alternative feed ingredients to reduce feed costs in the context of international maize price fluctuation. Rice is a staple cereal grain for human consumption. Currently, the self-sufficiency rate for rice is close to 100% in China. Due to the high yield production and poor taste quality, there is a serious backlog of early *indica* rice in China, especially in the southern rice-producing provinces. Thus, rice may have the potential to be used as an alternative energy source for pigs in the context of international maize price fluctuation. As a result of evaluating the effects of replacing corn with paddy rice in pig diets, using paddy rice in swine diets at up to 15% for growing pigs, 20% for early-finishing pigs, and 30% for late-finishing pigs did not cause adverse impacts on growth performance. Therefore, paddy rice can be evaluated as a valuable feed ingredient for swine diets during the growing–finishing period, if favorably priced.

**Abstract:**

Three experiments were conducted to evaluate paddy rice as an alternative energy feedstuff in low-protein diets for pigs. In Experiment 1, a total of 400 growing pigs (20.68 ± 0.29 kg initial bodyweight), were randomly allocated four dietary treatments with 0, 10, 15, and 20% paddy rice for 30 days. Feeding 10% or 15% paddy rice had no adverse impacts on average daily gain (ADG) and feed to gain ratio (F:G), while the inclusion of 20% rice in diets significantly influenced the growth performance of pigs. In Experiment 2, 364 early-finishing pigs (42.25 ± 0.47 kg) were divided into four treatments with 0, 15, 20, and 25% paddy rice for 35 days. Feeding 15% or 20% paddy rice had no negative consequences on growth performance, while pigs fed with 25% rice had the lowest ADG and the greatest F:G. In Experiment 3, 364 late-finishing pigs (79.52 ± 1.28 kg) were divided into four treatments with 0, 20, 25, and 30% paddy rice for 60 days. Paddy rice can be included at up to 30% in diets without compromising growth performance, while feeding with 25% rice significantly improved the performance for pigs compared with the corn-fed control.

## 1. Introduction

Swine diets contain amounts of cereal grains such as corn, wheat, and oat [1]. Given its widespread availability, corn is the primary ingredient served as an energy source in many parts of the world [2,3]. However, recently, corn prices and supply have been fluctuating due to high demand in industrial and energy sectors [4,5]. Thus, other cereal grains with stable supplies and cost-efficiencies may be considered as possible alternative ingredients for pigs. Similar to corn, rice is an important energy component, while the use of rice as feedstuff has been restricted in previous years [6,7]. Compared with corn, the price of rice remains relatively stable in China. In December 2022, the national price of rice was CNY 2717 per ton while the price of corn was CNY 2893 per ton according to the National Food and Strategic Reserves Administration (NFSRA) (http://www.lswz.gov.cn/html/zmhd/lysj/lsjg-scjc_3.shtml, accessed on 10 January 2024). Moreover, rice contains a wide variety of essential vitamins, minerals, phenolic compounds, and numerous volatile components, which are superior to those found in wheat and maize [8]. In addition, rice starch has a smaller structure and granule size, a unique flavor and taste, as well as a hypoallergenic property compared to other cereals [4,9]. China is the largest producer and consumer of rice in the world [10,11]. Currently, the self-sufficiency rate for rice is close to 100% in China [5,12]. With the development of agricultural technology, rice production has increased by more than 50% over the past 50 years [12]. The primary rice production in China for 2023 was 206.60 million tons, according to the National Bureau of Statistics of China (NBSC) (https://www.stats.gov.cn/sj/zxfb/202312/t20231211_1945417.html, accessed on 10 January 2024). Rice cultivated in China includes two main subspecies, *indica* rice and *japonica* rice [13]. Compered to *japonica* rice, *indica* rice produces higher yields and has a shorter growth duration [14]. However, *indica* rice has a lower gel consistency and a higher amylose content, which negatively affect its cooking and eating quality [13]. Due to high yield production and poor taste quality, there is a serious backlog of early *indica* rice in China [11]. As a substitution of maize, paddy rice has been widely used for feeding pigs and poultry in rural China, especially in the southern provinces [11]. Therefore, rice may have the potential to be used as alternative energy feedstuff for livestock in the context of international maize price fluctuation.

The nutritional composition of rice varies depending on the degree of grinding [4]. Previous studies have shown that both brown rice (unpolished) and white rice (polished) can be used as a major source of dietary carbohydrates without negatively affecting the nutrient digestibility and growth performance of pigs [4,6]. Given the higher cost and nutrient losses during the milling process, paddy rice seems more cost-effective than milled rice. However, paddy rice has a high fiber content, which may lead to negative impacts on growth performance. Thus, there is a need to evaluate the extent to which paddy rice could be used to replace corn in swine diets without adverse influence, especially when the basal diet is low in crude protein (CP) to reduce feed costs. The present study was designed to evaluate paddy rice as an alternative ingredient in substitution of corn for pigs fed on protein-restricted diets. Given the fact that the fiber digestibility of pigs varied with growth stages, three experiments were conducted to investigate the inclusion levels of rice for growing, early-finishing, and late-finishing pigs under commercial conditions.

## 2. Materials and Methods

### 2.1. The Experimental Design, Animals, and Diets

Three experiments were conducted on a commercial pig farm (Ganzhou, China). All pigs (Landrace × Yorkshire × Duroc) were housed in pens equipped with slatted floors, stainless-steel feeders and nipple waterers. Pens (similar in bodyweight, pig number, and sex distribution) were assigned to different dietary treatments in a randomized complete block design. Pigs had ad libitum access to feed and water throughout the trial. The paddy rice (*indica* variety) used in the current study was purchased from a commercial rice mill (Ganzhou, China) and ground to a small particle size (approximately 0.5 mm) using a 2 mm screen. All diets were fed in pellet form and formulated to contain all essential nutrients that met or exceeded the NRC (2012) recommendations for growing and finishing pigs [15]. The basal diet was based on corn and soybean meal, while the other experimental diets were formulated with different inclusion levels of paddy rice. All chemical analyses were performed according to AOAC (Association of Official Analytical Chemists, 2005) methods [16]. Prepared samples were measured in duplicate for dry matter (DM, Method 934.01), ether extract (EE, Method 920.39), crude fiber (CF, Method 985.29), crude ash (CA, Method 942.05), and starch (Method 979.10). Contents of neutral detergent fiber (NDF) and acid detergent fiber (ADF) were determined by Ankom 2000 Fiber Analyzer (Macedon, NY, USA). The nitrogen was examined using KDN-103 automatic kjeldahl nitrogen analyzer (Shanghai, China), and CP was calculated by multiplying the total nitrogen by 6.25 (Method 990.03). Moreover, the apparent total tract digestibility (ATTD) of nutrients was determined by chromium oxide indicator method and following the procedure described previously [17]. Gross energy (GE) was analyzed using a bomb calorimeter (MP-C 2000, Mingpeng Technology, Changsha, China), and chromium was examined using an absorption spectrophotometer (Biotek, Rochester, NY, USA). Ingredients and calculated and analyzed nutrient composition of diets were presented in Table 1, Table 2 and Table 3.

Three experiments were conducted to evaluate the effects of partially replacing corn with paddy rice on the growth performance of growing (Experiment 1, 20 to 40 kg bodyweight), early-finishing (Experiment 2, 40 to 80 kg), and late-finishing (Experiment 3, 80 to 140 kg) pigs under commercial conditions. Protein-restricted diets with different protein levels of 15.45%, 13.50%, 10.50% were utilized in Experiment 1, 2, and 3, respectively. In Experiment 1, a total of 400 growing pigs with an initial bodyweight of 20.68 ± 0.29 kg were assigned to 4 dietary treatments with 0, 10, 15, and 20% of paddy rice, respectively. There were 5 replicate pens with 20 pigs per pen. The feeding trail lasted for 30 days. In Experiment 2, a total of 364 early-finishing pigs (42.25 ± 0.47 kg) were allotted to 4 treatments with 7 replicate pens per treatment with 13 pigs per pen. Inclusion levels of rice were set at 0, 15, 20, and 25%. The pigs were fed the respective diets for 35 days. In Experiment 3, a total of 364 late-finishing pigs (79.52 ± 1.28 kg initial bodyweight) were allocated to 4 treatments formulated by containing 0, 20, 25 or 30% paddy rice. Each dietary treatment had 7 replicate pens with 13 pigs per pen. The feeding trial was conducted for 60 days.

### 2.2. Performance Evaluation and Statistical Analysis

In order to assess growth performance, pigs were weighed at the beginning and the end of the experiment. The amounts of feed intake were recorded. Average daily gain (ADG), average daily feed intake (ADFI), and the feed to gain ratio (F:G) were calculated. Data were examined by one-way ANOVA followed by Tukey’s post hoc analysis. Orthogonal polynomial contrast was used to evaluate the linear and quadratic effects of the inclusion level of rice on growth performance. Statistical significance was defined as *p*-value < 0.05, while the tendency was accepted at 0.05 < *p* < 0.1. The rice levels for optimal performance were determined by subjecting the ADG and F:G to the linear broken-line [y = L + U × (R − x), where (R − x) is zero when x > R], as described previously [18], using the NLIN procedure of SAS (version 9.4, SAS Inst. Inc., Cary, NC, USA).

## 3. Results

### 3.1. Experiment 1

For growing pigs of 20 to 40 kg bodyweight, final bodyweight (FBW) was significantly different among dietary groups (Table 4). Feeding with 10% or 15% paddy rice had no significant adverse effects on ADG and F:G, while pigs fed with 20% rice had lower ADG and greater F:G than those without the rice inclusion (*p* < 0.05). Moreover, both FBW and ADG showed a linear reduction while F:G increased linearly (*p* < 0.05) with increasing rice levels. No significant difference was observed for total feed intake (TFI) or ADFI, while a quadratic improvement was noticed as rice levels increased (*p* < 0.05). In addition, no significant difference was found for the ATTD of DM, GE, and CP among groups.

### 3.2. Experiment 2

For early-finishing pigs of 40 to 80 kg bodyweight, there was no significant difference with respect to FBW, TFI, or ADFI among groups (Table 5). Feeding 15% or 20% paddy rice had no negative consequences on ADG and F:G compared with those without the substitution, while pigs fed diets with 25% rice had the lowest ADG and the greatest F:G (*p* < 0.05). According to a polynomial contrast analysis, ADG was found to be decreased in a linear manner (*p* < 0.05), while F:G increased linearly (*p* < 0.05) with rice inclusion levels increasing. However, no significant difference was noted for the ATTD of DM, GE, and CP among groups.

### 3.3. Experiment 3

For late-finishing pigs of 80 to 140 kg bodyweight, a trend of increase in FBW was observed (*p* = 0.079), while TFI, ADFI, as well as the ATTD of DM, GE, and CP was not affected by increasing dietary rice levels (Table 6). Noticeably, pigs fed 25% rice diets had higher ADG and lower F:G than those fed un-supplemented diets or 20% rice diets (*p* < 0.05). Compared with control groups, pigs that received diets containing 30% rice had a numerical improvement in ADG and F:G, though it did not reach statistical significance. With increasing rice supply, ADG showed a linear increase (*p* < 0.05) while F:G tended to decrease linearly (*p* = 0.076). Furthermore, based on the linear analysis model, the inclusion levels of rice to maximize ADG and minimize F:G for late-finishing pigs were 25% and 26.27%, respectively (Figure 1A,B).

## 4. Discussion

To achieve economic benefits and cope with international maize price fluctuation, the potential of using paddy rice as an alternative ingredient in swine diets was assessed in the current study. For growing pigs in Experiment 1, inclusions of 10% or 15% paddy rice in diets had no detrimental effects on ADG and F:G, while feeding with 20% rice significantly influenced the growth performance of pigs. For early-finishing pigs in Experiment 2, feeding with paddy rice at up to 20% did not negatively affect growth performance, while the inclusion of 25% rice significantly hampered animal performance. The level of crude fiber (CF) in paddy rice (10.20%) used in this study was markedly higher than that in corn (2.10%). It has been reported that dietary fiber was necessary to promote digestive system development, promote intestinal peristalsis, and avoid gut ailments for animals, while fiber-rich ingredients might decrease ADFI and carcass yields [19]. However, the feed intake of pigs with or without paddy rice inclusions was not different in the present study, suggesting that palatability or feed preference might not be the main reason for the decreased growth performance in Experiments 1 and 2. The consumption of high levels of dietary fiber might influence the digestibility of nutrients [20]. In Experiments 1 and 2, there was a numerical decrease in the ATTD of DM, GE, and CP among groups, but this was without statistical significance. These findings were in line with a previous study showing that the addition of different fibrous ingredients had no effects on the ATTD of the DM or GE of growing pigs [21]. A possible reason for this observation may be the fact that the hindgut fermentation was enhanced as the concentration of dietary fiber increased [22]. The reduced productive performance may be in part due to the lower energetic efficiency of the fermented products [21]. Soluble dietary fiber has been shown to possess a high water-holding property, thereby retarding gastric emptying and slowing nutrient absorption rates, while insoluble dietary fiber has been found to bind organic compounds and increase fecal bulk [23]. Additionally, the use of fibrous diets may increase the thickness of the unstirred water layer adjacent to gut mucosa, leading to impaired nutrient utilization [24]. A previous study also demonstrated that a high fiber content might form a physical barrier to block enzymes’ access to starch granules to restrain starch hydrolysis [25]. Moreover, increased dietary fiber is able to adsorb amino acids (AAs) and peptides, withholding them from digestion, and hence limiting AAs’ availability for protein deposition in growing pigs [26]. Based on the above analysis, the observed adverse effects of paddy rice might be associated with a higher dietary fiber content, which may contribute to a reduced nutrient utilization and consequently result in lower pig performance.

For late-finishing pigs in Experiment 3, the inclusion of 25% rice in diets significantly improved productive performance, while feeding with up to 30% paddy rice had a numerical improvement compared with those without the substitution. Observations from some previous studies have indicated that the decline in the growth rate of pigs fed with high fiber diets was mostly limited to the growing phase rather than the finishing phase, while other research suggested that high-fiber diets had no detrimental impacts in the growing or finishing periods [24,27,28]. Findings from the present study were in agreement with a previous report, showing that the addition of dietary fiber to extruded rice-based diets did not hamper pig growth performance [29]. Consistent with our research, a previous study has reported that an increase in dietary fiber did not adversely affect the nutrient digestibility in growing–finishing pigs, when adjusting dietary CP content to the lowest possible values [19]. It has been hypothesized that pigs were able to tolerate a wide range of fiber levels when dietary energy was adequate in their diet [19,30]. Dietary fibers are mainly fermented in the hindgut of pigs, resulting in the production of volatile fatty acids (VFAs), and accordingly contributing to the dietary energy supply [31]. A recent study has revealed that substituting corn with brown rice could significantly change the fecal bacterial composition, with the increase in phylum *Bacteroidetes* and decrease in *Firmicutes* in finishing pigs [17]. Moreover, the small granule size of rice starch (approximately 2 μm) makes it easier to digest than corn [4,9], which may partially contribute to the improved performance observed in late-finishing pigs. Although no differences were observed for the carcass characteristics of pigs during the finishing period when replacing corn with brown rice [17], the effects of paddy rice on the carcass characteristics and meat quality of pigs still need further investigation.

## 5. Conclusions

Collectively, paddy rice can be used as alternative energy feedstuff in swine diets at up to 15% for growing pigs, 20% for early-finishing pigs, and 30% for late-finishing pigs without negative effects on growth performance. The biological mechanisms (e.g., changes in gut microbiota and the correlated metabolites) behind these observations need further exploration and explanation.

## Figures and Tables

**Figure 1 animals-14-00391-f001:**
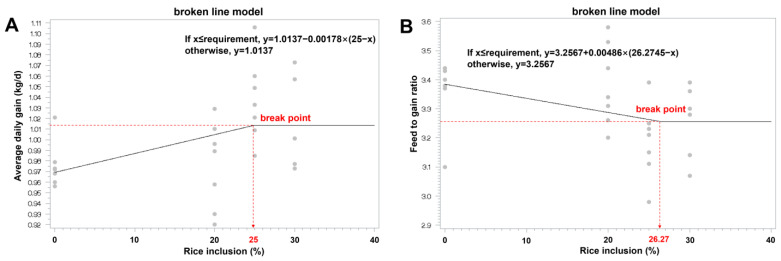
Estimation of optimal inclusion levels of paddy rice for late-finishing pigs. Fitted linear plot (black line) of ADG (**A**) and F:G (**B**) as a function. The optimal level determined by linear analysis for ADG was 25% (y plateau = 1.01), and for F:G was 26.27% (y plateau = 3.26). Each gray dot represented one replicate.

**Table 1 animals-14-00391-t001:** Ingredients and calculated and analyzed nutrient composition of the diets for growing pigs (Experiment 1).

	Inclusion Levels of Paddy Rice, %			
Item	0	10	15	20
**Ingredients, %**				
Corn ^#^	74.41	62.58	56.66	50.75
Paddy rice ^§^	-	10	15	20
Soybean meal (43.90% CP)	20.84	21.27	21.49	21.70
Soybean oil	0.42	1.86	2.57	3.29
Limestone	1.07	1.05	1.04	1.03
Mono-calcium and di-calcium phosphate	1.05	1.07	1.08	1.09
Salt	0.40	0.40	0.40	0.40
Choline chloride	0.07	0.07	0.07	0.07
Vitamin–mineral premix *	0.50	0.50	0.50	0.50
L-Lys sulfate 70%	0.70	0.68	0.67	0.66
L-Thr 98.50%	0.16	0.17	0.17	0.17
DL-Met 99%	0.17	0.17	0.17	0.17
L-Val 99%	0.14	0.13	0.13	0.13
L-Trp 98%	0.04	0.04	0.04	0.04
L-Ile 99%	0.03	0.02	0.02	0.02
Total	100	100	100	100
**Calculated composition**				
CP, %	15.45	15.45	15.45	15.45
NE, Mcal/kg	2.48	2.48	2.48	2.48
Starch, %	47.96	46.08	45.15	44.21
SID-Lys, %	1.02	1.02	1.02	1.02
SID-Met+Cys, %	0.58	0.58	0.58	0.58
SID-Met, %	0.37	0.37	0.37	0.37
SID-Thr, %	0.62	0.62	0.62	0.62
SID-Trp, %	0.18	0.18	0.18	0.18
SID-Val, %	0.67	0.67	0.67	0.67
SID-Ile, %	0.54	0.54	0.54	0.54
SID-Leu, %	1.19	1.15	1.14	1.12
SID-His, %	0.37	0.36	0.36	0.36
SID-Arg, %	0.84	0.86	0.88	0.89
SID-Phe, %	0.65	0.65	0.64	0.64
SID-Tyr, %	0.43	0.43	0.43	0.43
**Analyzed values, %**				
CP	15.83	15.74	15.55	15.62
EE	2.67	2.54	2.45	2.42
CF	2.94	3.74	4.14	4.54
CA	4.61	4.85	4.99	5.09

Abbreviations: SID = standardized ileal digestibility, CP = crude protein, NE = net energy, Lys = lysine, Met = methionine, Cys = cysteine, Thr = threonine, Trp = tryptophan, Val = valine, Ile = isoleucine, Leu = leucine, His = histidine, Arg = arginine, Phe = phenylalanine, Tyr = tyrosine, EE = ether extract, CF = crude fiber, CA = crude ash. ^#^ Nutritional values for corn used in the experiment: dry matter 86.09%, CP 7.34%, EE 3.10%, CF 2.10%, CA 1.18%, starch 63.92%, neutral detergent fiber 9.75%, acid detergent fiber 2.62%. ^§^ Paddy rice used in this study was *indica* subspecies and had the following analysis values: dry matter 87.40%, CP 7.06%, EE 1.90%, CF 10.20%, CA 3.62%, starch 56.80%, neutral detergent fiber 24.95%, acid detergent fiber 11.80%. * Premix provided the following per kilogram of complete diet: vitamin A, 9000 IU; vitamin D3, 2400 IU; vitamin E, 20 IU; vitamin K3, 3 mg; thiamine, 1.4 mg; riboflavin, 4 mg; pyridoxine, 3 mg; vitamin B12, 12 μg; nicotinic acid, 30 mg; pantothenic acid, 14 mg; folic acid, 0.8 mg; biotin, 44 μg; Fe, 76 mg; Cu, 240 mg; Zn, 76 mg; Mn, 20 mg; I, 0.48 mg; Se, 0.4 mg.

**Table 2 animals-14-00391-t002:** Ingredients and calculated and analyzed nutrient composition of the diets for early-finishing pigs (Experiment 2).

	Inclusion Levels of Paddy Rice, %			
Item	0	15	20	25
**Ingredients, %**				
Corn	79.42	62.94	57.03	51.11
Paddy rice	-	15	20	25
Soybean meal (43.90% CP)	15.42	16.31	16.53	16.74
Soybean oil	-	1.92	2.64	3.36
Wheat bran	1.29	-	-	-
Limestone	1.01	0.97	0.96	0.95
Mono-calcium and di-calcium phosphate	0.85	0.91	0.92	0.93
Salt	0.40	0.40	0.40	0.40
Choline chloride	0.07	0.07	0.07	0.07
Vitamin–mineral premix *	0.50	0.50	0.50	0.50
L-Lys sulfate 70%	0.63	0.59	0.58	0.57
L-Thr 98.50%	0.14	0.14	0.14	0.14
DL-Met 99%	0.12	0.12	0.12	0.12
L-Val 99%	0.10	0.08	0.08	0.08
L-Trp 98%	0.04	0.04	0.03	0.03
L-Ile 99%	0.02	0.01	0.01	0.01
Total	100	100	100	100
**Calculated composition**				
CP, %	13.50	13.50	13.50	13.50
NE, Mcal/kg	2.48	2.48	2.48	2.48
Starch, %	51.35	49.06	48.12	47.19
SID-Lys, %	0.86	0.86	0.86	0.86
SID-Met+Cys, %	0.49	0.49	0.49	0.49
SID-Met, %	0.29	0.30	0.30	0.30
SID-Thr, %	0.54	0.54	0.54	0.54
SID-Trp, %	0.16	0.16	0.16	0.16
SID-Val, %	0.56	0.56	0.56	0.56
SID-Ile, %	0.46	0.46	0.46	0.46
SID-Leu, %	1.07	1.03	1.01	1.00
SID-His, %	0.32	0.32	0.31	0.31
SID-Arg, %	0.70	0.74	0.76	0.77
SID-Phe, %	0.56	0.56	0.56	0.56
SID-Tyr, %	0.37	0.37	0.38	0.38
**Analyzed values, %**				
CP	13.74	13.63	13.69	13.62
EE	2.92	2.62	2.54	2.47
CF	3.04	4.14	4.51	4.88
CA	4.19	4.52	4.66	4.76

Abbreviations: SID = standardized ileal digestibility, CP = crude protein, NE = net energy, Lys = lysine, Met = methionine, Cys = cysteine, Thr = threonine, Trp = tryptophan, Val = valine, Ile = isoleucine, Leu = leucine, His = histidine, Arg = arginine, Phe = phenylalanine, Tyr = tyrosine, EE = ether extract, CF = crude fiber, CA = crude ash. * Premix provided the following per kilogram of complete diet: vitamin A, 9000 IU; vitamin D3, 2400 IU; vitamin E, 20 IU; vitamin K3, 3 mg; thiamine, 1.4 mg; riboflavin, 4 mg; pyridoxine, 3 mg; vitamin B12, 12 μg; nicotinic acid, 30 mg; pantothenic acid, 14 mg; folic acid, 0.8 mg; biotin, 44 μg; Fe, 76 mg; Cu, 240 mg; Zn, 76 mg; Mn, 20 mg; I, 0.48 mg; Se, 0.4 mg.

**Table 3 animals-14-00391-t003:** Ingredients and calculated and analyzed nutrient composition of the diets for late-finishing pigs (Experiment 3).

	Inclusion Levels of Paddy Rice, %			
Item	0	20	25	30
**Ingredients, %**				
Corn	83.76	66.40	60.48	54.57
Paddy rice	-	20	25	30
Soybean meal (43.90% CP)	6.27	8.34	8.55	8.77
Soybean oil	-	1.73	2.44	3.16
Wheat bran	6.43	-	-	-
Limestone	0.71	0.53	0.52	0.50
Mono-calcium and di-calcium phosphate	0.94	1.22	1.23	1.25
Salt	0.40	0.40	0.40	0.40
Choline chloride	0.07	0.07	0.07	0.07
Vitamin–mineral premix *	0.50	0.50	0.50	0.50
L-Lys sulfate 70%	0.60	0.53	0.52	0.51
L-Thr 98.50%	0.12	0.11	0.11	0.12
DL-Met 99%	0.05	0.07	0.07	0.07
L-Val 99%	0.08	0.06	0.05	0.05
L-Trp 98%	0.04	0.04	0.03	0.03
L-Ile 99%	0.04	0.02	0.02	0.01
Total	100	100	100	100
**Calculated composition**				
CP, %	10.50	10.50	10.50	10.50
NE, Mcal/kg	2.48	2.48	2.48	2.48
Starch, %	55.11	53.96	53.02	52.09
SID-Lys, %	0.65	0.65	0.65	0.65
SID-Met+Cys, %	0.38	0.38	0.38	0.38
SID-Met, %	0.20	0.22	0.22	0.22
SID-Thr, %	0.41	0.41	0.41	0.41
SID-Trp, %	0.12	0.12	0.12	0.12
SID-Val, %	0.43	0.43	0.43	0.43
SID-Ile, %	0.34	0.34	0.34	0.34
SID-Leu, %	0.87	0.85	0.83	0.81
SID-His, %	0.25	0.24	0.24	0.24
SID-Arg, %	0.49	0.55	0.56	0.57
SID-Phe, %	0.42	0.43	0.43	0.43
SID-Tyr, %	0.28	0.29	0.29	0.29
**Analyzed values, %**				
CP	10.78	10.65	10.71	10.62
EE	3.16	2.76	2.65	2.52
CF	2.83	4.11	4.58	4.94
CA	3.54	3.96	4.05	4.19

Abbreviations: SID = standardized ileal digestibility, CP = crude protein, NE = net energy, Lys = lysine, Met = methionine, Cys = cysteine, Thr = threonine, Trp = tryptophan, Val = valine, Ile = isoleucine, Leu = leucine, His = histidine, Arg = arginine, Phe = phenylalanine, Tyr = tyrosine, EE = ether extract, CF = crude fiber, CA = crude ash. * Premix provided the following per kilogram of complete diet: vitamin A, 6000 IU; vitamin D3, 2400 IU; vitamin E, 20 IU; vitamin K3, 2 mg; thiamine, 0.96 mg; riboflavin, 5.3 mg; pyridoxine, 2 mg; vitamin B12, 12 μg; nicotinic acid, 22 mg; pantothenic acid, 11.2 mg; folic acid, 0.4 mg; biotin, 40 μg; Fe, 76 mg; Cu, 120 mg; Zn, 76 mg; Mn, 12 mg; I, 0.24 mg; Se, 0.4 mg.

**Table 4 animals-14-00391-t004:** Effect of feeding paddy rice on the growth performance and apparent total tract digestibility of growing pigs of 20 to 40 kg.

Item	Paddy Rice	%			SEM ^1^		*p* Value	
	0	10	15	20		ANOVA	Linear	Quadratic
Initial bodyweight, kg	20.66	20.83	20.47	20.78	0.06	0.215	0.997	0.572
Final bodyweight, kg	39.82	39.72	39.10	38.99	0.13	0.038	0.007	0.979
Total feed intake, kg	36.98	37.14	37.22	36.93	0.05	0.196	0.921	0.042
Average daily gain, kg	0.64 ^a^	0.63 ^ab^	0.62 ^ab^	0.61 ^b^	0.00	0.010	0.001	0.662
Average daily feed intake, kg	1.23	1.24	1.24	1.23	0.00	0.195	0.928	0.042
Feed to gain ratio	1.93 ^b^	1.97 ^ab^	2.00 ^ab^	2.03 ^a^	0.01	0.014	0.002	0.942
Dry matter, %	84.90	83.22	83.63	83.06	0.52	0.621	0.254	0.677
Gross energy, %	83.98	83.20	81.83	79.96	0.89	0.428	0.128	0.557
Crude protein, %	77.99	76.44	76.07	75.73	0.53	0.470	0.130	0.753

^1^ SEM = standard error of mean. In each row, values with different superscript letters are significantly different (*p* < 0.05).

**Table 5 animals-14-00391-t005:** Effect of feeding paddy rice on the growth performance and apparent total tract digestibility of early-finishing pigs of 40 to 80 kg.

Item	Paddy Rice	%			SEM		*p* Value	
	0	15	20	25		ANOVA	Linear	Quadratic
Initial bodyweight, kg	41.92	42.21	42.37	42.47	0.09	0.135	0.021	0.905
Final bodyweight, kg	79.86	79.90	79.04	78.57	0.24	0.141	0.044	0.300
Total feed intake, kg	80.38	80.63	81.18	81.17	0.24	0.565	0.190	0.955
Average daily gain, kg	1.08 ^a^	1.08 ^ab^	1.05 ^ab^	1.03 ^b^	0.01	0.030	0.006	0.321
Average daily feed intake, kg	2.30	2.30	2.32	2.32	0.01	0.567	0.190	0.868
Feed to gain ratio	2.12 ^b^	2.14 ^ab^	2.22 ^ab^	2.25 ^a^	0.02	0.014	0.002	0.367
Dry matter, %	86.27	84.35	84.48	84.11	0.71	0.726	0.289	0.767
Gross energy, %	84.07	82.50	82.54	81.25	0.76	0.661	0.239	0.870
Crude protein, %	82.48	82.07	81.81	81.81	0.71	0.988	0.731	0.981

In each row, values with different superscript letters are significantly different (*p* < 0.05).

**Table 6 animals-14-00391-t006:** Effect of feeding paddy rice on the growth performance and apparent total tract digestibility of late-finishing pigs of 80 to 140 kg.

Item	Paddy Rice	%			SEM		*p* Value	
	0	20	25	30		ANOVA	Linear	Quadratic
Initial bodyweight, kg	79.70	80.04	79.19	79.14	0.24	0.521	0.423	0.322
Final bodyweight, kg	138.23	138.59	141.44	139.41	0.49	0.079	0.111	0.894
Total feed intake, kg	196.87	197.49	198.29	197.32	0.38	0.635	0.415	0.587
Average daily gain, kg	0.98 ^b^	0.98 ^b^	1.04 ^a^	1.00 ^ab^	0.01	0.015	0.038	0.671
Average daily feed intake, kg	3.28	3.29	3.30	3.29	0.01	0.680	0.477	0.612
Feed to gain ratio	3.37 ^ab^	3.38 ^a^	3.19 ^b^	3.28 ^ab^	0.03	0.039	0.076	0.531
Dry matter, %	84.91	86.20	86.45	85.81	0.58	0.825	0.459	0.611
Gross energy, %	84.20	85.41	86.85	85.88	0.55	0.412	0.157	0.783
Crude protein, %	82.27	82.91	84.35	83.70	0.64	0.715	0.343	0.971

In each row, values with different superscript letters are significantly different (*p* < 0.05).

## Data Availability

All data will be available from the corresponding author upon reasonable request.

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
