# Peer review of "An Evaluation of Paddy Rice as an Alternative Energy Source in Protein-Restricted Diets for Growing, Early-Finishing, and Late-Finishing Pigs"

_animals, 2024, doi:10.3390/ani14030391_

Round 1

Reviewer 1 Report

Comments and Suggestions for Authors

This article presents interesting productive information for swine production, but its scientific scope is limited. It only presents productive data, it should add some information, for example, how the digestibility of nutrients is modified with the use of paddy rice. In this way, the article would have much more scientific clarity helping in its use.

Specific comments:

Line 67: Clarifying the specific NRC guidelines or standards referenced, and how the relate to the study, will help readers understand the basis of your nutritional assessments and decisions. This detail is crucial for ensuring clarity and aiding in the reproducibility of your research.

Line 70: when referencing AOAC in your document, it would be beneficial to mention what AOAC means and the specific AOAC methods used for feed analysis and why they were chosen.

Line 130: it would be beneficial to include a more detailed explanation of the NLIN procedure from SAS.

Line 197, 199: It is suggested to delete “-“ in high-fiber. (could be simplified to high fiber)

Line 200, 208: When citing sources in your document, it´s not necessary to use italics when citing directly in the text. For example, in a sentence like “In agreement with Kim et al. (2008), who found…” “et al.” is not italicized. This distinction helps maintain clarity and adheres to common academic citation practices. I recommend reviewing your entire document.

Materials and methods

While it´s mentioned that feed samples were analyzed according to AOAC methods, specifying key methods used for nutrient analysis would be beneficial. Overall, this section is well-written, providing essential details for understanding and replicating the study.

Results

This section is well-structured and presents the data clearly, with comprehensive tables and appropriate statistical analysis.

Discussion

It aptly compares the study´s findings with existing research, enhancing the credibility and context of the results.

Conclusion

It effectively encapsulates the main outcomes of the experiments, clearly stating the acceptable inclusion levels of paddy rice in pig diets at different growth stages. This section is concise and to the point, making it easy for readers to grasp the overall findings of the study.

It would be beneficial to suggest specific areas or questions for future research in biological mechanisms behind the observations.

Author Response

Comments and Suggestions for Authors

This article presents interesting productive information for swine production, but its scientific scope is limited. It only presents productive data, it should add some information, for example, how the digestibility of nutrients is modified with the use of paddy rice. In this way, the article would have much more scientific clarity helping in its use.

Response: We appreciate the reviewer’s positive evaluation of our work. The apparent total tract digestibility of nutrients has been determined by chromium oxide indicator method while no significant changes were observed in our study. We have added this information to Supplementary Materials section (Supplementary Table S1) according to your valuable advice. Thanks again for your insightful comments.

Specific comments:

Line 67: Clarifying the specific NRC guidelines or standards referenced, and how the relate to the study, will help readers understand the basis of your nutritional assessments and decisions. This detail is crucial for ensuring clarity and aiding in the reproducibility of your research.

Response: We are very grateful for your careful work and comments. We have revised the Materials and Methods section as suggested.

Line 70: when referencing AOAC in your document, it would be beneficial to mention what AOAC means and the specific AOAC methods used for feed analysis and why they were chosen.

Response: We are grateful for the suggestion and we have described the specific AOAC methods as suggested, on page 2 and 3, lines 90-101.

Line 130: it would be beneficial to include a more detailed explanation of the NLIN procedure from SAS.

Response: Thank you for your suggestion. We have included a detailed NLIN procedure according to your valuable advice, on page 6, lines 160-161.

Line 197, 199: It is suggested to delete “-” in high-fiber. (could be simplified to high fiber).

Response: Thank you for your careful review. We have deleted “-” accordingly, on page 9, line 229.

Line 200, 208: When citing sources in your document, it´s not necessary to use italics when citing directly in the text. For example, in a sentence like “In agreement with Kim et al. (2008), who found…” “et al.” is not italicized. This distinction helps maintain clarity and adheres to common academic citation practices. I recommend reviewing your entire document.

Response: Thank you very much for your careful review. We have revised the manuscript accordingly, on page 9, line 233 and 240.

Materials and methods

While it´s mentioned that feed samples were analyzed according to AOAC methods, specifying key methods used for nutrient analysis would be beneficial. Overall, this section is well-written, providing essential details for understanding and replicating the study.

Response: We are grateful for the helpful suggestion and we have described the specific AOAC methods as suggested, on page 2 and 3, lines 90-101.

Results

This section is well-structured and presents the data clearly, with comprehensive tables and appropriate statistical analysis.

Response: Thank you for your nice comments on our article.

Discussion

It aptly compares the study´s findings with existing research, enhancing the credibility and context of the results.

Response: We appreciate the reviewer’s positive evaluation of our work.

Conclusion

It effectively encapsulates the main outcomes of the experiments, clearly stating the acceptable inclusion levels of paddy rice in pig diets at different growth stages. This section is concise and to the point, making it easy for readers to grasp the overall findings of the study.

It would be beneficial to suggest specific areas or questions for future research in biological mechanisms behind the observations.

Response: Thank you very much for your careful review and constructive suggestions. We have revised the Conclusion section as suggested, on page 9, lines 252-253.

Reviewer 2 Report

Comments and Suggestions for Authors

This manuscript explores the viability of using paddy rice as a substitute for maize in swine diets across different growth stages. The study is well-structured with three distinct experiments, each targeting a different growth stage of pigs. This approach provides a comprehensive understanding of the impact of paddy rice across the entire lifecycle of swine. The study addresses a problem—fluctuations in international maize prices—by proposing an alternative that can reduce feed costs while maintaining growth performance.

While the paper discusses the effects of paddy rice, it doesn’t provide a detailed comparative analysis with other potential feed substitutes. Such a comparison could have strengthened the case for paddy rice.

The study, while detailed, doesn’t address the health effects or meat quality implications of feeding pigs with paddy rice, which are critical for end consumers. This needs to be mentioned in the discussion.

Line 43-44: This sentence lacks a reference. “early indica rice in China”, Was this the kind rice used in this study?

Because rice is produced in China as food for people, its price will be higher than those of crops as animal feed. Although the international corn price fluctuates, it is worth explaining whether it is cost-effective and feasible to feed pigs with rice in the Introduction. The necessity and significance of this study need to be clarified.

Line 81: Was paddy rice crushed? What was the particle size? Was the feed granular or powdery? Where Was the experimental rice produced? Was it old rice? This information is of interest to readers.

Figure 1. Estimation of optimal inclusion levels of paddy rice for late-finishing pigs. These plots are good, but why not fit linear plots of ADG and F:G as a function for early-finishing pigs (40-80kg) and growing pigs(20-40kg)?

Overall, the manuscript provides valuable insights into the use of paddy rice as an alternative feed ingredient.

Author Response

Comments and Suggestions for Authors

This manuscript explores the viability of using paddy rice as a substitute for maize in swine diets across different growth stages. The study is well-structured with three distinct experiments, each targeting a different growth stage of pigs. This approach provides a comprehensive understanding of the impact of paddy rice across the entire lifecycle of swine. The study addresses a problem—fluctuations in international maize prices—by proposing an alternative that can reduce feed costs while maintaining growth performance.

Response: We appreciate the reviewer’s positive evaluation of our work.

While the paper discusses the effects of paddy rice, it doesn’t provide a detailed comparative analysis with other potential feed substitutes. Such a comparison could have strengthened the case for paddy rice.

Response: We are very grateful for your careful work and thoughtful suggestions. We have revised the Introduction section as suggested, on page 2, lines 45-48. “Moreover, rice contains a wide variety of essential vitamins, minerals, phenolic com-pounds, and numerous volatile components, which are superior to those found in wheat and maize [8]. In addition, rice starch has smaller structure and granule size, unique flavor and taste, as well as hypoallergenic property compared with other cereals [4, 9].”

The study, while detailed, doesn’t address the health effects or meat quality implications of feeding pigs with paddy rice, which are critical for end consumers. This needs to be mentioned in the discussion.

Response: Thank you very much for your careful review and insightful comments. We have revised the Discussion section according to your valuable advice, on page 9, lines 245-248.

Line 43-44: This sentence lacks a reference. “early indica rice in China”, Was this the kind rice used in this study?

Response: Thank you for your careful review. The indica variety was used in our study, and we have added this information on page 2, line 84. And we have added the reference accordingly, on page 2, lines 60-61. “Liu XJ, Guo JH, Xue L, Zhao D, Liu G. Where has all the rice gone in China? A farm-to-fork material flow analysis of rice supply chain with uncertainty analysis. Resour Conserv Recy. 2023; 190:106853.”

Because rice is produced in China as food for people, its price will be higher than those of crops as animal feed. Although the international corn price fluctuates, it is worth explaining whether it is cost-effective and feasible to feed pigs with rice in the Introduction. The necessity and significance of this study need to be clarified.

Response: We are grateful for the suggestion and we have revised the Introduction Section according to your valuable advice, on pages 1-2, lines 40-61.

Line 81: Was paddy rice crushed? What was the particle size? Was the feed granular or powdery? Where Was the experimental rice produced? Was it old rice? This information is of interest to readers.

Response: Thank you very much for your careful review and constructive advice. The rice used in our study was new and purchased from a commercial rice mill and ground to a small particle size (approximately 0.5 mm) using a 2 mm screen. All diets were fed in pellet form. We have added this information to the Materials and Methods section as suggested, on page 2, lines 83-86.

Figure 1. Estimation of optimal inclusion levels of paddy rice for late-finishing pigs. These plots are good, but why not fit linear plots of ADG and F:G as a function for early-finishing pigs (40-80kg) and growing pigs(20-40kg)?

Response: Thank you for your careful review. We have also conducted the NLIN procedure for growing and early-finishing pigs. However, no breakpoints were observed and therefore we did not fit linear plots for these pigs.

Overall, the manuscript provides valuable insights into the use of paddy rice as an alternative feed ingredient.

Response: Thank you again for your positive comments and valuable suggestions to improve the quality of our manuscript.

Round 2

Reviewer 1 Report

Comments and Suggestions for Authors

The authors made most of the suggested changes. However, the main one related to nutrient digestibility was partly considered. They must incorporate it as part of the body of the article, not as supplementary material, so that the paper gains scientific power. In the same way, they should discuss those results, and how they relate to the productive results described.

Author Response

ID:animals-2816544

Dear Editor,

Thanks a lot for having reviewed our manuscript. Now we have revised the manuscript according to the comments. We believe that we have now adequately addressed all the concerns raised in the review and have strengthened the overall quality of our manuscript. The changes that we have made to the manuscript are marked using track changes. Our point-by-point response to the review follows.

1#

Comments and Suggestions for Authors

The authors made most of the suggested changes. However, the main one related to nutrient digestibility was partly considered. They must incorporate it as part of the body of the article, not as supplementary material, so that the paper gains scientific power. In the same way, they should discuss those results, and how they relate to the productive results described.

Response: We are very grateful for your work and thoughtful suggestions. We have incorporated the nutrient digestibility into Table 4, 5 and 6 according to your helpful advice. We have also revised the Result section (on page 7, lines 173-174, 185-186, 191) and Discussion section (on page 9, lines 220-228) accordingly and 3 more relevant literatures have been cited in the text. Thanks again for your insightful comments.
